# Impact of Chemotherapy Dose Intensity on Pathological Complete Response in Pembrolizumab-Treated Early Triple-Negative Breast Cancer: A Real-World Multicenter Analysis

**DOI:** 10.3390/cancers17213554

**Published:** 2025-11-02

**Authors:** Palma Fedele, Stefania Luigia Stucci, Matteo Landriscina, Maria Morritti, Francesco Giuliani, Lucia Moraca, Giuseppe Cairo, Raffaele Ardito, Marianna Giampaglia, Domenico Bilancia, Assunta Melaccio, Antonella Terenzio, Antonio Gnoni, Antonella Licchetta, Federica Fumai, Laura Lanotte, Alessandro Rizzo, Gennaro Gadaleta-Caldarola

**Affiliations:** 1Oncology Unit, Dario Camberlingo Hospital, Francavilla Fontana, 72021 Brindisi, Italy; coro.francavilla@asl.brindisi.it; 2Breast Care Unit, University Hospital Consortium Policlinico of Bari, 70124 Bari, Italy; stuccistefania@gmail.com; 3U.O. Medical Oncology and Biomolecular Therapy, Department of Medical and Surgical Sciences, University of Foggia, 71100 Foggia, Italy; matteo.landriscina@unifg.it (M.L.); antonella.terenzio@unifg.it (A.T.); 4Oncology Unit, Fondazione “Casa Sollievo della Sofferenza”, IRCCS, San Giovanni Rotondo, 71013 Foggia, Italy; maria.morritti@gmail.com; 5Oncology Unit, San Paolo Hospital, 70123 Bari, Italy; francesco.giuliani@asl.bari.it (F.G.); assunta.melaccio@asl.bari.it (A.M.); 6Oncology Unit, Teresa Masselli Mascia Hospital, San Severo, 71013 Foggia, Italy; lucia.moraca@aslfg.it; 7Oncology Unit, Vito Fazzi Hospital, 73100 Lecce, Italy; giuseppecairo15@gmail.com; 8Day Hospital Oncologico IRCCS CROB, Rionero in Vulture, 85028 Potenza, Italy; raffaele.ardito@crob.it; 9Oncology Unit, San Carlo Hospital, 85100 Potenza, Italy; mariannagiampaglia@yahoo.it (M.G.); domenicobilancia@gmail.com (D.B.); 10Oncology Unit, Sacro Cuore di Gesù Hospital, Gallipoli, 73014 Lecce, Italy; antonio.gnoni@asl.le.it (A.G.); antonella.licchetta@asl.le.it (A.L.); 11Oncology Unit, Mons. Dimiccoli Hospital, 70051 Barletta, Italy; laura.lanotte@aslbat.it (L.L.); gennaro.gadaleta@aslbat.it (G.G.-C.); 12S.S.D. C.O.r.O. Bed Management Presa in Carico, TDM, IRCCS Istituto Tumori “Giovanni Paolo II”, 70124 Bari, Italy; rizzo.alessandro179@gmail.com

**Keywords:** triple-negative breast cancer, pembrolizumab, pathological complete response, dose intensity, real-world study

## Abstract

**Simple Summary:**

Triple-negative breast cancer (TNBC) is an aggressive disease where pembrolizumab combined with chemotherapy has improved the likelihood of achieving a pathological complete response (pCR). Yet, in daily practice, many patients require dose reductions or discontinuations, and little is known about their real-world impact. In this large multicenter Italian study, we not only confirmed the relevance of maintaining chemotherapy dose intensity but also analyzed pregnancy history and comorbidities, two variables rarely explored in previous series. Our results highlight the importance of supportive measures to preserve treatment delivery and suggest novel directions for future biomarker research.

**Abstract:**

**Background:** Pembrolizumab combined with neoadjuvant chemotherapy significantly improves pCR in early TNBC, but the effect of treatment intensity and baseline clinical factors has been insufficiently explored in real-world settings. **Methods:** We retrospectively included 169 consecutive patients with stage II–III TNBC treated across 11 Italian oncology centers (January 2022–January 2025) with the KEYNOTE-522 regimen. Clinical, pathological, and treatment data were collected, including relative dose intensity (RDI), dose modifications, and toxicities. The primary endpoint was pCR (ypT0/is ypN0). **Results:** The overall pCR rate was 65.7%, which is consistent with clinical trial data. Dose reductions occurred in 40% of patients and chemotherapy was discontinued in 18%. Patients maintaining RDI ≥85% achieved higher pCR (79.3% vs. 51.2%, *p* < 0.001). Similarly, patients without dose reductions (72.5% vs. 55.2%, *p* = 0.031) and those completing all cycles (73.1% vs. 41.0%, *p* < 0.001) had superior outcomes. Dose modifications occurred mainly during the taxane/carboplatin phase and were predominantly due to hematological toxicities (anemia 44%, neutropenia 30%, and thrombocytopenia 15%), neuropathy (18%), and gastrointestinal events (36%). Higher TILs correlated with increased pCR (70.6% vs. 60.7%, *p* = 0.049), while BRCA mutations showed a favorable trend. ECOG, BMI, pregnancy history, and comorbidities were not significantly associated with pCR. **Conclusions:** In this multicenter real-world cohort, maintaining chemotherapy dose intensity (RDI ≥ 85%) and completing all planned cycles were strongly associated with higher pCR rates, reinforcing the clinical importance of minimizing dose reductions and discontinuations during pembrolizumab-based neoadjuvant therapy for TNBC.

## 1. Introduction

Triple-negative breast cancer (TNBC) represents approximately 15–20% of all breast cancers and is characterized by an aggressive clinical course, early recurrence, and limited therapeutic options [1,2]. Neoadjuvant chemotherapy (NACT) with anthracyclines and taxanes has long represented the standard of care, with pathological complete response (pCR) being a well-recognized surrogate for long-term outcomes [3]. Patients failing to achieve pCR after NACT have a substantially worse prognosis, underscoring the prognostic value of this endpoint in both clinical trials and real-world practice [3].

The phase III KEYNOTE-522 trial established the addition of pembrolizumab-to-platinum-based NACT as a new standard in early TNBC, demonstrating a significant improvement in pathological complete response (64.8% vs. 51.2% with placebo) and event-free survival (EFS) [4]. Updated analyses have recently shown that pembrolizumab-based therapy also provides a statistically significant improvement in overall survival (OS), further reinforcing the long-term impact of this approach [5]. The introduction of immune checkpoint inhibitors into the perioperative setting has therefore redefined treatment paradigms, marking a shift from pure chemotherapy dependence toward integrated chemo-immunotherapy. This transition highlights the need to optimize both treatment intensity and patient selection to preserve efficacy while minimizing toxicity.

Despite the robust evidence provided by clinical trials, real-world data remain essential to understand how pembrolizumab-based regimens perform outside strictly controlled settings. In daily practice, dose reductions or early discontinuations of chemotherapy are frequent due to cumulative toxicity, hematologic adverse events, or comorbid conditions. Such modifications may compromise treatment efficacy, particularly when dose intensity falls below critical thresholds. The concept of relative dose intensity (RDI) has been extensively validated in adjuvant breast cancer, where maintaining ≥85% of planned dose intensity correlates with improved disease-free and overall survival [6,7,8]. However, its relevance in the context of chemo-immunotherapy for early TNBC remains underexplored.

Beyond treatment-related variables, several patient- and tumor-specific factors may influence pCR. Younger age has been associated with higher pCR rates, potentially reflecting enhanced immune reactivity and fewer comorbidities [9]. Conversely, obesity and elevated BMI have been linked to systemic inflammation and reduced treatment sensitivity [10]. Performance status (ECOG), pregnancy history, and pre-existing autoimmune disorders may further modulate the host immune environment, while genomic alterations such as BRCA mutations and immune biomarkers like PD-L1 expression or tumor-infiltrating lymphocytes (TILs) can also affect response [11,12]. Understanding how these factors interact with treatment delivery in real-world populations is crucial for tailoring patient management.

Furthermore, while the prognostic role of pCR is well established, the optimal management of patients achieving pCR remains an evolving field. Ongoing studies such as OPTIMICE-pCR [13] aim to refine immunotherapy duration and intensity, exploring de-escalation strategies for patients with excellent responses. In parallel, real-world evidence can complement these efforts by identifying clinical predictors of response and quantifying the impact of treatment adherence on outcomes.

In this context, the present multicenter Italian study sought to evaluate the association between chemotherapy dose intensity and pCR in patients with early TNBC treated with pembrolizumab-based NACT. By incorporating patient-related variables, such as pregnancy history and comorbidities, and tumor-related markers including TILs and BRCA status, we aimed to provide a comprehensive picture of the clinical factors that influence treatment delivery and response in everyday oncology practice.

## 2. Materials and Methods

We retrospectively included 169 consecutive patients with early TNBC diagnosed and treated at 11 Italian oncology centers between January 2022 and January 2025. Eligible patients had histologically confirmed TNBC, stage II–III disease at diagnosis, and received pembrolizumab in combination with neoadjuvant chemotherapy according to the KEYNOTE-522 regimen, consisting of four cycles of paclitaxel (80 mg/m^2^ weekly) plus carboplatin (AUC 5 every 3 weeks or AUC 1.5 weekly), followed by four cycles of doxorubicin (60 mg/m^2^) or epirubicin (90 mg/m^2^) combined with cyclophosphamide (600 mg/m^2^) every 3 weeks. Patients presenting with metastatic disease, prior systemic therapy for breast cancer, or incomplete clinical or pathological data were excluded.

Clinical and pathological information was retrieved from electronic medical records at each participating center and anonymized in a centralized database. Collected variables included demographic and clinical features (age at diagnosis, menopausal status, body mass index, ECOG performance status, history of pregnancies, and comorbidities); tumor characteristics (clinical stage, histological grade, Ki-67, tumor-infiltrating lymphocytes, PD-L1 status, and BRCA mutation status); and treatment details (number of cycles planned and delivered, dose reductions, discontinuations and their causes, treatment delays, relative dose intensity, and immune-related adverse events graded according to CTCAE v4.0) [14]). The use of granulocyte colony-stimulating factor (G-CSF), administered either prophylactically or therapeutically according to the treating physician’s discretion, was also recorded. In most cases, G-CSF support was given during the taxane/carboplatin phase to mitigate the risk of myelosuppression.

Relative dose intensity (RDI) was calculated for each patient as the ratio of the delivered dose intensity to the planned dose intensity, expressed as a percentage. Dose intensity was defined as the amount of drug (mg/m^2^) administered per week. For patients who received multiple agents, RDI was averaged across all drugs included in the regimen. A threshold of 85% was used to categorize patients into high (≥85%) and low (<85%) RDI groups. This threshold is consistent with prior oncology literature, where it has been established as the optimal cut-off for early-stage breast cancer regimens [6], supported by meta-analyses showing improved survival outcomes with RDI ≥ 80–85% [7], and recognized in treatment intensity studies as the level beyond which no additional benefit is gained [8].

The primary outcome of interest was pathological complete response (pCR), defined as the absence of invasive disease in both breast and lymph nodes (ypT0/is ypN0).

The primary objective of the study was to assess the association between chemotherapy dose modifications and the likelihood of achieving pCR. Secondary objectives included describing pCR rates according to tumor biomarkers (such as TILs and BRCA status), determining the frequency and causes of chemotherapy discontinuation, and identifying baseline patient characteristics (including ECOG performance status, body mass index, pregnancy history, and comorbidities) associated with dose modifications. PD-L1 expression was not included in the analyses, as this biomarker is not required for pembrolizumab use in the early TNBC setting and was not consistently available in the participating centers’ records. Categorical variables were summarized as absolute frequencies and percentages, while continuous variables were reported as median and interquartile range. Group comparisons were performed using χ^2^ or Fisher’s exact test for categorical variables, and Student’s *t*-test or Mann–Whitney U test for continuous variables, as appropriate based on data distribution and expected frequencies. Logistic regression models were applied to explore predictors of pCR and chemotherapy dose modifications. All statistical tests were two-sided, and a *p*-value < 0.05 was considered statistically significant. Analyses were performed using SPSS version 29 (IBM Corp., Armonk, NY, USA).

The study was conducted in accordance with the Declaration of Helsinki and was approved by the *Comitato Etico Territoriale Regione Puglia—Azienda Ospedaliero-Universitaria “Consorziale Policlinico”* (study code 7889).

Written informed consent was obtained from patients whenever required by local regulations governing retrospective studies.

## 3. Results

A total of 169 patients were included in the analysis. The median age at diagnosis was 53.3 years (range 47.2–58.6), and the median BMI was 25.2 (range 22.5–28.4). Most patients had an ECOG performance status of 0 (81.7%). BRCA mutations were detected in 10.7% of cases, and the median TILs value was 25.8%, with a nearly equal distribution between high (50.3%) and low (49.7%) TILs. A history of pregnancy was reported in 74.0% of patients, while comorbidities were present in 46.2%, most commonly thyroid disorders (11.2%) and autoimmune/inflammatory conditions (6.5%). Other comorbidities, including hypertension, gastrointestinal, hepatic, cardiovascular, respiratory, renal, and psychiatric diseases, were less frequent. (Table 1).

The overall pathological complete response (pCR) rate was 65.7% (111/169). Chemotherapy dose modifications were common: 40% of patients experienced dose reductions and 18% discontinued chemotherapy, although pembrolizumab was generally continued according to schedule. Relative dose intensity (RDI) was calculated for the overall chemotherapy backbone rather than for each single drug, to reflect the cumulative treatment exposure. Pembrolizumab was administered at a fixed dose and according to schedule in almost all patients, with discontinuation occurring in only 10% of cases; therefore, no meaningful RDI-based analysis was feasible for this agent.

The likelihood of achieving pCR was significantly affected by treatment intensity. Patients who did not require dose reduction achieved a pCR rate of 72.5%, compared with 55.2% among those with reductions (*p* = 0.031). The effect was even more pronounced for discontinuation: patients who completed chemotherapy achieved a pCR rate of 73.1%, while those who discontinued had a pCR of only 41.0% (*p* < 0.001). Similarly, relative dose intensity (RDI) was strongly correlated with outcome: patients with RDI ≥ 85% had a pCR rate of 79.3%, compared with 51.2% in those with RDI < 85% (*p* < 0.001).

Dose modifications occurred more frequently during the taxane/carboplatin phase, which accounted for approximately two-thirds of dose reductions (65%) and about 70% of discontinuations. The anthracycline/cyclophosphamide phase was less frequently affected. Hematological toxicities were the leading cause of modifications, followed by neuropathy and gastrointestinal events. Anemia was the most common hematologic toxicity (44%), followed by neutropenia (30%, with grade 3–4 events in only 1.2%) and thrombocytopenia (21.9%). Nausea (36%) and peripheral neuropathy (18%) were the most frequent non-hematologic events. Immune-related adverse events were less common but still clinically relevant (Table 2)

In exploratory analyses, baseline characteristics were not significantly associated with pCR. Patients younger than 50 years had a pCR rate of 66.1% compared with 65.3% among those ≥50 years (*p* = 0.902). Similarly, pCR was achieved in 66.0% of patients with ECOG 0 and in 64.5% with ECOG 1 (*p* = 0.791), and in 66.4% of patients with BMI < 25 compared with 65.0% in those with BMI ≥ 25 (*p* = 0.822). Pregnancy history also did not influence outcomes: 61.9% of nulliparous patients achieved pCR compared with 47.2% of those with at least one pregnancy (*p* = 0.099). The presence of comorbidities showed no impact, with pCR rates of 49.5% in patients without comorbidities and 53.8% in those with comorbidities (*p* = 0.434). Younger patients, however, were more likely to discontinue chemotherapy, while ECOG status, BMI, pregnancy history, and comorbidities were not significantly associated with treatment modifications (Table 3).

To further assess whether the association between chemotherapy dose intensity and pCR was independent of clinical and biological covariates, a multivariate logistic regression analysis was performed, including age, ECOG performance status, BMI, BRCA mutation, and TILs (Table 4). In the adjusted model, treatment intensity did not independently predict the likelihood of achieving pCR (OR = 1.18, 95% CI 0.67–2.05, *p* = 0.562). Similarly, age (OR = 1.05, 95% CI 0.75–1.46, *p* = 0.852), BMI (OR = 1.10, 95% CI 0.80–1.50, *p* = 0.751), ECOG ≥ 1 (OR = 1.02, 95% CI 0.60–1.75, *p* = 1.000), BRCA mutation (OR = 1.08, 95% CI 0.58–2.02, *p* = 0.876), and TILs (per 10%, OR = 1.05, 95% CI 0.86–1.28, *p* = 1.000) were not significantly associated with pCR. These findings indicate that, in this cohort, none of the analyzed clinical or biological variables independently predicted pCR after adjustment for the covariates considered. These findings suggest that, in this cohort, no single clinical or biological covariate emerged as an independent predictor of pCR after adjustment for the variables considered.

Pathological complete response outcomes according to treatment intensity and toxicity patterns are summarized in Figure 1, highlighting the superior pCR rates observed among patients maintaining full-dose chemotherapy delivery

## 4. Discussion

The integration of pembrolizumab into the neoadjuvant treatment of TNBC has led to significant improvements in pathological complete response (pCR), as first demonstrated in the KEYNOTE-522 trial, where the addition of immune checkpoint blockade to standard chemotherapy increased pCR rates from 51.2% to 64.8% [15]. The overall pCR rate in our cohort (65.7%) was consistent with these pivotal findings, confirming the effectiveness of this regimen in a real-world, multicenter Italian population.

In line with observations from clinical trials and retrospective series [16,17], we demonstrated that dose reductions, discontinuations, and relative dose intensity (RDI) were strongly associated with treatment outcomes. Patients maintaining an RDI ≥ 85% achieved a markedly higher pCR rate (79.3%) compared with those below this threshold (51.2%, *p* < 0.001). Although the 85% cut-off is widely used across hematology and oncology to define adequate chemotherapy delivery [6,7,8], few real-world reports in the immunotherapy setting have confirmed its prognostic impact. Our results, therefore, provide novel evidence that preserving dose intensity remains critical even in the era of chemo-immunotherapy.

These findings are consistent with emerging real-world evidence. Buonaiuto et al. [18] recently reported that higher RDI of neoadjuvant chemotherapy in TNBC was associated with improved pCR and event-free survival, confirming the clinical relevance of the 85% threshold also in unselected populations. Similarly, Mezzanotte-Sharpe et al. [19] showed that dose modifications during chemo-immunotherapy correlated with a lower likelihood of achieving pCR, further reinforcing the negative prognostic impact of treatment intensity reduction in routine practice. Together with our results, these analyses highlight the importance of supportive strategies to maintain chemotherapy delivery and preserve outcomes in TNBC patients treated with pembrolizumab-based regimens.

In comparison with other real-world studies investigating dose intensity in the neoadjuvant treatment of TNBC [11,16,18,19,20], our analysis is distinctive for including a relatively large multicenter Italian cohort, thereby enhancing the representativeness of its findings.

Regarding treatment modifications, we observed that approximately two-thirds of dose reductions and 70% of discontinuations occurred during the taxane/carboplatin phase, with hematological toxicities being the leading cause. This finding is consistent with prior studies, which highlighted the increased myelosuppression associated with platinum-containing regimens [16,21]. Peripheral neuropathy and gastrointestinal adverse events also contributed significantly to treatment modifications, while immune-related adverse events (irAEs) were less frequent but not negligible. Compared with trial data, where grade ≥ 3 anemia and thrombocytopenia occurred in approximately 6–7% of patients [15,16], our results confirm that severe anemia and thrombocytopenia remain the most clinically relevant safety concerns in routine practice.

Exploratory analyses of baseline characteristics further enriched our understanding of treatment response. Consistent with prior reports [22,23], higher levels of tumor-infiltrating lymphocytes (TILs) were associated with increased likelihood of pCR (70.6% vs. 60.7%).

In addition, we investigated pregnancy history, a variable not previously addressed in neoadjuvant immunotherapy trials. In our series, pregnancy history was not significantly associated with response: pCR was achieved in 61.9% of nulliparous patients compared with 47.2% of those with at least one pregnancy (*p* = 0.099). Although no systematic analyses are available in TNBC, there is a strong biological rationale for considering pregnancy as a potential modifier of immune response [24]. Pregnancy induces profound immunological adaptations, including maternal–fetal tolerance, persistence of fetal microchimeric cells, and antigen sharing between placenta and tumor tissue. These mechanisms may exert long-term effects on immune surveillance and responsiveness to immune checkpoint inhibitors. Evidence from other tumor types suggests gender-related differences in immunotherapy efficacy, with women deriving less benefit from PD-1 monotherapy but equal or greater benefit from chemo-immunotherapy combinations [25,26]. Within this context, our results represent an exploratory but novel contribution that could stimulate future research into pregnancy-related immunological markers as predictors of immunotherapy response.

A further limitation of our study is the lack of detailed information on the interval between the last pregnancy and cancer diagnosis. The biological and immunological consequences of pregnancy may vary considerably depending on whether breast cancer occurs shortly after delivery—when postpartum involution and transient immune adaptations are still active—or many years later. Future prospective studies should incorporate this temporal variable to better elucidate the relationship between pregnancy-related immune modulation and immunotherapy outcomes.

Despite its strengths, this study has limitations. Its retrospective design may have introduced selection bias and limited the completeness of clinical data. Although the cohort represents one of the largest multicenter real-world series in Italy, the sample size remains relatively small for subgroup analyses, particularly for pregnancy history and comorbidities, and follow-up is not yet sufficient to assess survival outcomes. In addition, data were collected across multiple institutions, which may have introduced variability in clinical management and reporting. Finally, some exploratory findings should be interpreted with caution, given the absence of comprehensive multivariable adjustment.

While pCR is an established surrogate endpoint for long-term outcomes in TNBC, its prognostic value in real-world settings may vary depending on patient selection and treatment intensity. Our cohort is currently under longitudinal follow-up, and future analyses will assess whether pCR correlates with event-free and overall survival, thereby confirming its validity as a surrogate marker in this population.

## 5. Conclusions

In conclusion, our real-world experience confirms the efficacy and safety of the KEYNOTE-522 regimen in an unselected population, while highlighting the prognostic relevance of treatment intensity and the impact of dose modifications. These findings underscore the need for supportive strategies to preserve chemotherapy delivery and for further research to identify patient-related biomarkers that can refine treatment selection and optimization.

## Figures and Tables

**Figure 1 cancers-17-03554-f001:**
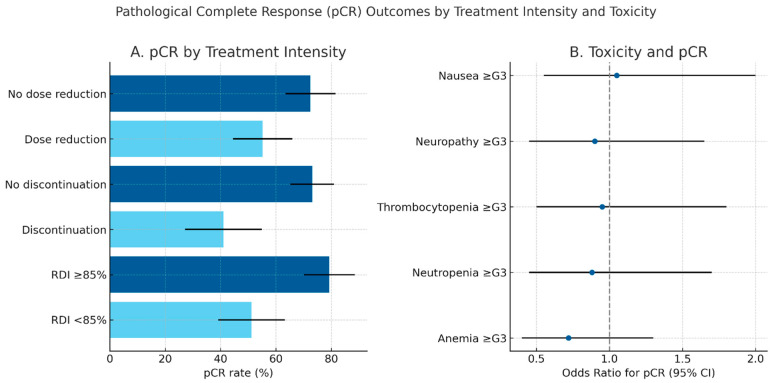
Pathological complete response (pCR) outcomes by treatment intensity and toxicity. (**A**) Bar plot showing pCR rates according to chemotherapy dose reduction, discontinuation, and relative dose intensity (RDI). Error bars indicate 95% confidence intervals. Maintaining full-dose intensity (RDI ≥ 85%) and completing all planned chemotherapy cycles were associated with higher pCR rates. (**B**) Forest plot illustrating the relationship between major grade ≥ 3 toxicities and the probability of achieving pCR, expressed as odds ratios (OR) with 95% confidence intervals. Hematological adverse events (anemia, neutropenia, thrombocytopenia) were the most frequent, but no individual toxicity emerged as significantly associated with pCR.

**Table 1 cancers-17-03554-t001:** Baseline characteristics of the study population.

Variable	Value
Number of patients	169
Median age (years)	53.3 (47.2–58.6)
Median BMI (kg/m^2^)	25.2 (range 22.5–28.4)
ECOG 0	138 (81.7%)
ECOG ≥ 1	31 (18.3%)
Mutated BRCA	18 (10.7%)
Wild-type BRCA	151 (89.3%)
Median TILs (%)	25.8
High TILs	85 (50.3%)
Low TILs	84 (49.7%)
≥1 pregnancy	125/169 (74.0%)
Any comorbidities	78/169 (46.2%)
• Thyroid disease	19/169 (11.2%)
• Autoimmune/inflammatory	11/169 (6.5%)
• Hypertension	4/169 (2.4%)
• Gastrointestinal	4/169 (2.4%)
• Hepatic disease	2/169 (1.2%)
• Cardiovascular disease	1/169 (0.6%)
• Respiratory disease	1/169 (0.6%)
• Renal disease	1/169 (0.6%)
• Psychiatric	1/169 (0.6%)

**Table 2 cancers-17-03554-t002:** Treatment-related adverse events according to CTCAE v4.0.

Adverse Event	Grade 1–2 *n* (%)	Grade 3–4 *n* (%)	Any Grade *n* (%)
Neutropenia	49 (28.9%)	2 (1.2%)	51 (30.2%)
Anemia	64 (37.9%)	10 (6.1%)	74 (44.0%)
Thrombocytopenia	27 (16.0%)	10 (5.9%)	37 (21.9%)
Elevated alanine aminotransferase level	21 (12.4%)	4 (2.4%)	25 (14.8%)
Vomiting	24 (14.2%)	3 (1.8%)	27 (16.0%)
Nausea	58 (34.3%)	3 (1.8%)	61 (36.1%)
Peripheral neuropathy	27 (16.0%)	3 (1.8%)	30 (17.8%)
Myalgia	11 (6.5%)	18 (10.7%)	29 (17.2%)
Fatigue/Asthenia	3 (1.8%)	1 (0.6%)	4 (2.4%)
Endocrinopathy (irAE)	2 (1.2%)	0 (0.0%)	2 (1.2%)
Rash (irAE)	24 (14.2%)	3 (1.8%)	27 (16.0%)

Notes: Percentages are calculated on the overall study population (*n* = 169). Toxicities are reported according to CTCAE v4.0. (irAE) = immune-related adverse event.

**Table 3 cancers-17-03554-t003:** Pathological complete response (pCR) according to treatment modifications and biomarkers.

Variable	Category	N	pCR *n*/N (%)	*p*-Value
Dose reduction	No	102	74/102 (72.5%)	0.031
	Yes	67	37/67 (55.2%)	
Discontinuation	No	130	95/130 (73.1%)	<0.001
	Yes	39	16/39 (41.0%)	
RDI	<85%	82	42/82 (51.2%)	<0.001
	≥85%	87	69/87 (79.3%)	
TILs	Low	84	51/84 (60.7%)	0.049
	High	85	60/85 (70.6%)	
BRCA	Wild-type	151	98/151 (64.9%)	0.506
	Mutated	18	13/18 (72.2%)	
ECOG	0	138	91/138 (65.9%)	0.902
	≥1	31	20/31 (64.5%)	
BMI	<median	85	55/85 (64.7%)	0.791
	≥median	84	56/84 (66.7%)	
Pregnancy history	No	42	26/42 (61.9%)	0.099
	≥1	125	59/125 (47.2%)	
Any comorbidity	No	91	45/91 (49.5%)	0.434
	Yes	78	42/78 (53.8%)	

Notes: Percentages are row-wise. *p*-values from χ^2^ test or Fisher’s exact test, as appropriate (see Methods). pCR defined as ypT0/is ypN0. RDI threshold ≥ 85% as specified in Methods. TILs dichotomized by median (25.8%). BMI split at cohort median. Pregnancy history classified as ≥1 vs. none. Comorbidity defined as the presence of any concomitant disease reported at baseline.

**Table 4 cancers-17-03554-t004:** Multivariate logistic regression analysis of predictors of pathological complete response (pCR). The model included relative dose intensity (RDI ≥ 85%), age, ECOG performance status, body mass index (BMI), BRCA mutation status, and tumor-infiltrating lymphocytes (TILs). Results are expressed as odds ratios (OR) with 95% confidence intervals (CI).

	Variable	OR	CI 2.5%	CI 97.5%	*p*-Value
Intensity	No dose reduction (yes vs. no)	1.18	0.67	2.05	0.562
Age	per year	1.05	0.75	1.46	0.852
ECOG	ECOG ≥ 1 (vs. 0)	1.02	0.60	1.75	1.000
BMI	per unit	1.10	0.80	1.50	0.751
BRCA_mut	BRCA mutation (yes vs. no)	1.08	0.58	2.02	0.876
TILs_10	per 10% increase	1.05	0.86	1.28	1.000

## Data Availability

Due to the retrospective design and anonymization, the data are not shared publicly to protect patient privacy.

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
