# Peer review of "Impact of Chemotherapy Dose Intensity on Pathological Complete Response in Pembrolizumab-Treated Early Triple-Negative Breast Cancer: A Real-World Multicenter Analysis"

_cancers, 2025, doi:10.3390/cancers17213554_

Round 1
Reviewer 1 Report
Comments and Suggestions for Authors
Statistical Analysis
- Recommendation: Perform and report a multivariate logistic regression for pCR, adjusting for potential confounders such as age, ECOG status, BMI, BRCA mutation, and TILs.
This would provide stronger evidence that dose intensity independently predicts pCR.
Novelty
- The discussion could be strengthened by elaborating on how pregnancy-related immune adaptations (e.g., maternal immune tolerance mechanisms) might influence checkpoint inhibitor efficacy.
pCR as Endpoint
- Authors should acknowledge that while pCR is an accepted surrogate, long-term follow-up is needed to confirm its correlation with survival outcomes in this cohort.
Presentation
-
Consider summarizing pCR outcomes by treatment intensity and toxicity in graphical form (e.g., bar or forest plots).
-
Include confidence intervals alongside p-values for all key comparisons.
PD-L1 Status
- Suggest reporting descriptive statistics if PD-L1 was available for a subset, and briefly discussing potential bias introduced by its omission.
Author Response
Author's Reply to Reviewer 1
Comment 1
Statistical Analysis
-
Recommendation: Perform and report a multivariate logistic regression for pCR, adjusting for potential confounders such as age, ECOG status, BMI, BRCA mutation, and TILs.
This would provide stronger evidence that dose intensity independently predicts pCR.
Response 1
We thank the Reviewer for this constructive suggestion.
In accordance with the recommendation, we performed a multivariate logistic regression analysis including the following covariates: age, ECOG performance status, BMI, BRCA mutation status, and TILs, in addition to treatment intensity.
The results of this analysis have now been added to the Results section (Lines 224–260) and summarized in the new Table 4.
All modifications have been highlighted in yellow within the revised manuscript for your convenience.
Comment 2
Novelty
-
The discussion could be strengthened by elaborating on how pregnancy-related immune adaptations (e.g., maternal immune tolerance mechanisms) might influence checkpoint inhibitor efficacy.
Response 2
We thank the Reviewer for this thoughtful and constructive suggestion.
In accordance with the comment, we have expanded the Discussion section to include a paragraph describing how pregnancy-related immune adaptations—such as maternal–fetal tolerance, persistence of fetal microchimeric cells, and antigen sharing between placental and tumor tissue—may modulate immune surveillance and responsiveness to immune checkpoint inhibitors (line 313-322)
This addition highlights the potential long-term immunological imprint of pregnancy on antitumor immunity and provides a biological rationale for the exploratory evaluation of pregnancy history in our cohort.
All changes are highlighted in yellow in the revised manuscript for your review.
Comment 3
pCR as Endpoint
Authors should acknowledge that while pCR is an accepted surrogate, long-term follow-up is needed to confirm its correlation with survival outcomes in this cohort.
Response 3
We thank the Reviewer for this important observation.
We agree that, although pathological complete response (pCR) is a widely accepted surrogate endpoint for improved survival in triple-negative breast cancer, long-term follow-up is essential to validate its prognostic impact in our real-world cohort.
Accordingly, we have added a clarifying statement in the Discussion section (Lines 334–338) acknowledging this limitation and the need for future survival analyses.
Comment 4
Presentation
Consider summarizing pCR outcomes by treatment intensity and toxicity in graphical form (e.g., bar or forest plots).
Include confidence intervals alongside p-values for all key comparisons.
Response 4
As requested, a graphical summary of pCR outcomes by treatment intensity and toxicity has been added (Figure 1). Corresponding text describing the figure has been incorporated in the Results section (lines 254-261) to improve clarity and data visualization.
Comment 5
PD-L1 Status
-
Suggest reporting descriptive statistics if PD-L1 was available for a subset, and briefly discussing potential bias introduced by its omission.
Response 5
We thank the Reviewer for this valuable comment. PD-L1 expression was not consistently available across participating centers and was therefore excluded from the analyses. This choice reflects real-world clinical practice, as PD-L1 testing is not mandatory for pembrolizumab use in the neoadjuvant treatment of early triple-negative breast cancer. We have clarified this point in the Materials and Methods section (lines 152-155).
Reviewer 2 Report
Comments and Suggestions for Authors
In this work, the authors describe their findings in a real-world multicenter cohort of patients with TNBC that were treated with pembrolizumab combined with neoadjuvant chemotherapy. The authors highlight the importance of maintaining the chemotherapy dose intensity for a high pathological complete response. The authors also explore the association of other variables, such as TILs, pregnancy or patient comorbidities, with the response to the treatment. These studies are important because they analyze the results of the treatment proposed in clinical trials in a group of real-world patients. In the present manuscript, the results are well presented, and the conclusions are relevant for the clinical practice.
Author Response
Author's Reply to Reviewer 2
Comment 1
In this work, the authors describe their findings in a real-world multicenter cohort of patients with TNBC that were treated with pembrolizumab combined with neoadjuvant chemotherapy. The authors highlight the importance of maintaining the chemotherapy dose intensity for a high pathological complete response. The authors also explore the association of other variables, such as TILs, pregnancy or patient comorbidities, with the response to the treatment. These studies are important because they analyze the results of the treatment proposed in clinical trials in a group of real-world patients. In the present manuscript, the results are well presented, and the conclusions are relevant for the clinical practice.
Response 1
We sincerely thank the Reviewer for the positive evaluation of our work and for recognizing the relevance of our findings for clinical practice. We appreciate the constructive summary of the study’s strengths and have carefully addressed all specific comments provided below.
Reviewer 3 Report
Comments and Suggestions for Authors
-
Thank you for the opportunity to review this manuscript by Dr. Palma Fedele et al. The authors investigate the relationship between chemotherapy dose intensity and pathologic pCR. Although this is a potentially valuable study, several points require clarification and further detail.
-
The authors should provide detailed information on the preoperative chemotherapy regimens, including the number of cycles, doses of each agent, including anthracycline phase.
-
The relationship between pCR and the RDI of each drug, including pembrolizumab, should be analyzed and presented if data are available.
-
In the Discussion section, the authors state that “Pregnancy induces profound immunological adaptations, including maternal–fetal tolerance, persistence of fetal microchimeric cells, and antigen sharing between placenta and tumor tissue.” An appropriate reference should be cited to support this statement.
-
Author Response
Author's Reply to Reviewer 3
Comment 1
The authors should provide detailed information on the preoperative chemotherapy regimens, including the number of cycles, doses of each agent, including anthracycline phase.
Response 1
We thank the Reviewer for this helpful comment. All patients received neoadjuvant chemotherapy according to the KEYNOTE-522 regimen, which included four cycles of paclitaxel (80 mg/m² weekly) plus carboplatin (AUC 5 every 3 weeks or AUC 1.5 weekly) followed by four cycles of doxorubicin (60 mg/m²) or epirubicin (90 mg/m²) combined with cyclophosphamide (600 mg/m²) every 3 weeks. This information has been added to the Materials and Methods section (lines 114–118) for greater clarity.
Comment 2
The relationship between pCR and the RDI of each drug, including pembrolizumab, should be analyzed and presented if data are available.
Response 2
In our study, relative dose intensity (RDI) was calculated for the overall chemotherapy backbone rather than for individual agents, in order to capture the cumulative effect of dose reductions across the regimen. This approach reflects the design of prior dose-intensity analyses in early breast cancer and avoids overestimating the contribution of single drugs when multiple agents are administered sequentially.
Pembrolizumab was administered at a fixed dose according to schedule in nearly all patients, with discontinuation occurring in only 10% of cases, most often due to immune-related toxicity. Given the minimal variability in pembrolizumab dosing, no meaningful RDI-based analysis could be performed for this agent. These clarifications have been added to the Results section (lines 182-186).
Comment 3
In the Discussion section, the authors state that “Pregnancy induces profound immunological adaptations, including maternal–fetal tolerance, persistence of fetal microchimeric cells, and antigen sharing between placenta and tumor tissue.” An appropriate reference should be cited to support this statement.
Response 3:
We thank the Reviewer for this helpful suggestion. As recommended, we have now added the citation n 24 to support this statement (Colamatteo A, Fusco C, Micillo T, D’Hooghe T, de Candia P, Alviggi C, Longobardi S, Matarese G. Immunobiology of pregnancy: from basic science to translational medicine. Immunobiology. 2023;29(9):711–725. doi:10.1016/j.molmed.2023.05.009). The reference has been included in the Discussion section (line 315).